# Mental chronometry in big noisy data

**Edmund Wascher\*, Fariba Sharifian, Marie Gutberlet, Daniel Schneider, Stephan Getzmann, Stefan Arnau**

Dept. Ergonomics, IfADo–Leibniz Research Centre for Working Environment and Human Factors, Dortmund, Germany

\* wascher@ifado.de

## Abstract

Temporal measures (latencies) in the event-related potentials of the EEG (ERPs) are a valuable tool for estimating the timing of mental processes, one which takes full advantage of the high temporal resolution of the EEG. Especially in larger scale studies using a multitude of individual EEG-based tasks, the quality of latency measures often suffers from high and low frequency noise residuals due to the resulting low trial counts (because of compressed tasks) and because of the limited feasibility of visual inspection of the large-scale data. In the present study, we systematically evaluated two different approaches to latency estimation (peak latencies and fractional area latencies) with respect to their data quality and the application of noise reduction by jackknifing methods. Additionally, we tested the recently introduced method of Standardized Measurement Error (SME) to prune the dataset. We demonstrate that fractional area latency in pruned and jackknifed data may amplify within-subjects effect sizes dramatically in the analyzed data set. Between-subjects effects were less affected by the applied procedures, but remained stable regardless of procedure.

**Data Availability Statement:** https://osf.io/73f8u/.

**Funding:** The author(s) received no specific funding for this work.

## Introduction

Event-related potentials of the EEG (ERPs) have a long tradition when investigating temporal aspects of cognitive processing, which is also referred to as mental chronometry [1–3]. ERPs allow to estimate the timing of cognitive processes with high temporal resolution and thereby go far beyond what can be learned from behavior alone. The detection and parameterization of distinct time points in the signal (e.g. peak latencies, the time stamp of component-maxima in ERPs), however, requires a high-quality signal that is not always given. High and low frequency noise may overlay the signal of interest and compromise the reliable measurement of component latencies [4, 5].

In large scale studies, specific properties of the study design as well as requirements regarding data handling often hamper the analysis of data. To test a wide range of cognitive functions and behaviors in a reasonable amount of time, a battery of cognitive tests is usually presented as in the the "human connectome project", a series of studies that intends to investigate neural connections in the brain [6], which limits the time available for performing each single experimental task. The number of trials consequently is reduced compared to studies in which only a single task is employed. Trial counts, however, are a core factor of ERP data quality. In

**Competing interests:** The authors have declared that no competing interests exist.

addition, interactive and visually guided preprocessing of data (either for artifact rejection or manual correction of latency measures or deficient data sets) is often impractical and detrimental to the replicability when a large number of data sets have to be considered.

Thus, data processing needs to be standardized and well describable [7, 8]. In fact, if such large datasets need to be processed, it is recommended that all processing steps are completely automated [9, 10]. With respect to preprocessing pipelines, a number of solutions have been proposed that substantially increase data quality [9, 11, 12]. With respect to the quality of different latency measures, however, no systematic evaluation in large scale data sets has been provided thus far. In the present study, we compared the two main approaches to latency measurement in a sample of more than 500 participants, namely peak latency, and fractional area latency, both of which were applied to the data sets with additional approaches for noise reduction (see below). We tested the data obtained for reliability, internal consistency, but also with respect to effect sizes for specific experimental factors.

The measurement of peak latencies has a long tradition and still is the most commonly used method for the estimation of the timing of ERP components. Peak latency reflects the moment of the absolute amplitude maximum (positive or negative) in a predefined time window (see Fig 1). The time window is usually selected based on the grand average by visual inspection and the channel selected is most times derived from literature or from the actual topographic distribution of the component chosen. Data, in particular data with high frequency noise, often have more than one local maximum in this time window and the latency of the largest peak might not necessarily correspond to the center of the underlying ERP component [4, 5, 13]. As demonstrated in Fig 1, the measurement of fractional area latencies is more robust against high frequency noise [5, 14]. In this case, the time point within a restricted time window at which the accumulated area under the curve exceeds a defined portion (e.g. 50%) of the overall area is determined. This way, the influence of high frequency noise is minimized, however, for the handling of low frequency noise some further adaptions might be necessary [4].

Nevertheless, when a low number of trials contributes to the average in single subjects, the signal might still be very noisy which also makes the fractional area latency method prone to biases or distortions. A well-established approach to overcome this problem is the so-called jackknifing procedure [15, 16]. For this method, N averages are created that include N-1 subject averages each (leave-one-out averages). This way, a data set is created that is largely noise free, due to the large number of individual averages included, while still containing all of the interindividual variance. In particular in large datasets, the signal to noise ration raises massively this way, because of the decrease of the noise portion ($Noise_{estimated} = \frac{1}{\sqrt{N_{subj}*avgN_{trials}}}$). Measures derived from waveshapes that are generated by the jackknife method, will therefore be highly reliable. However, since the jackknifing procedure by its very nature reduces interindividual variance, statistics have to be adjusted to not overestimate experimental effects [17]. The most common approaches to correct jackknife statistics have not been worked out for all statistical analyses so far [18]. This particularly holds true when between-subject effects are considered with varying subjects counts within groups. Smulders [18] proposed a method to restore interindividual variance for measures that are derived from jackknifed data which should make the application of this method possible without further adjustments since any measured latency is set in relation to the entire data set (see Formula 1 from Smulders [18]).

$$Lat_{retrieved} = sum(Lat_{all}) - (n-1) * Lat_{measured}$$

So far, jackknifing has most often been applied in research on the Lateralized Readiness Potential (LRP), where the timing of motor processes is the central measure [17, 19, 20].

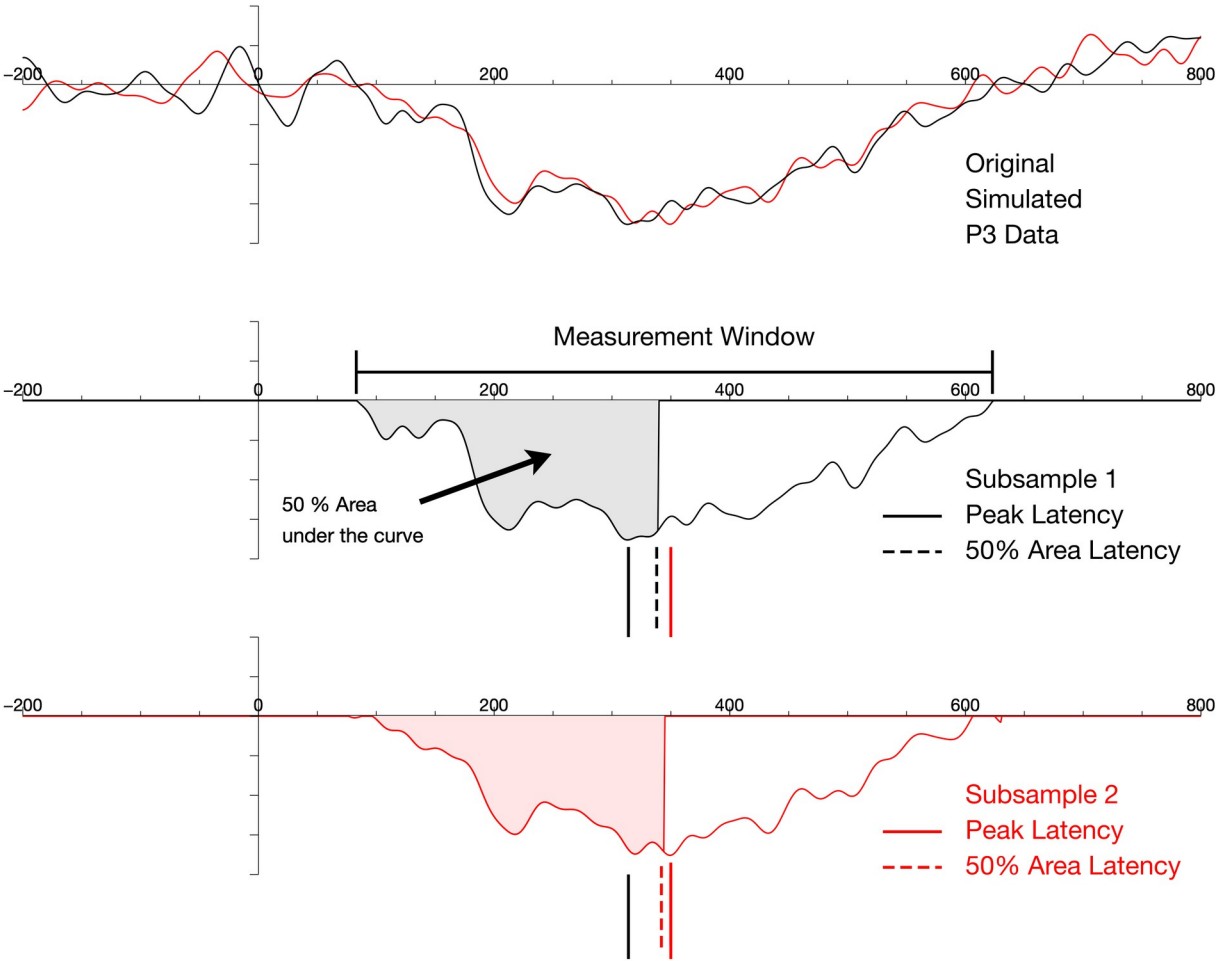

**Fig 1. Simulated data for two subsets of a typical posterior waveshape (black and red lines), reflecting a P3 component.** Both, peak latencies and 50% fractional area latencies were extracted. Peak latencies denote a single moment of maximal voltage within the designated measurement window. Fractional area latency denotes the moment when the cumulative area under the curve exceeds a defined criterion, here 50% of the overall area. Despite the high similarity of morphology across the two subsets, peak latencies would show an effect of more than 50 ms due to high frequency noise (solid vertical lines), that did not affect fractional area latency to the same amount (dashed vertical lines).

However, it might well be an approach that can be used for any kind of latency measure in the EEG. A simulation study, Kiesel and co-workers [21] demonstrated that jackknifing clearly outperforms measurements in individual subject averages but does not work for all types of parameters. In particular the peak picking method appears to show reduced power in jack-knifed data when compared to single subject averages, especially for slow components such as the P3 [21].

Using all of these approaches and considering the potential pitfalls, we compared different latency measures in a large dataset based on completely automated algorithms and tried to demonstrate their potential advantages and shortcomings. The dataset was collected in a modi-fied Psychomotor Vigilance Task [22]. In this task, participants had to perform a simple button press to occasionally appearing light flashes (stimulus onset asynchronies [SOA] 2, 3, 5, and 8 seconds). The task lasted, as in the original version, only 10 minutes. Thus, overall, only 132 stimuli were presented (33 trials for each of the 4 SOAs). Despite the short duration and the low number of stimuli presented, the PVT is, at least for behavioral data, often analyzed in 5 successive segments in order to demonstrate the decline of vigilance over time, the so-called

time on task (ToT) effect. Thus, the trial count for individual averages per conditions would never be larger than 26, which is determined by the truncated division of the number of overall trials (132) by the number of segments (5). This may be at the lower limit for some of the ERP-components of interest. Although the large sample size may help to obtain reliable latency differences despite the low trial count [23], small effects could be missed even in such a large data set.

ERP components of interest were the standard components that are prominent in visual tasks and that are typically elicited as part of the normal response to a visual stimulus. The early P1 and N1 components at posterior sites typically peak around 50 to 100 ms, and 100 to 200 ms after stimulus onset, respectively. Both components are evoked within the visual cortices and are assumed to be indicators of early sensory and attentional processing of incoming information [24, 25]. The P2 following the N1 peak at around 200 to 300 ms with a maximum over posterior areas. The P2 has been widely studied in relation to visual search and attention, with its amplitude being modulated by characteristics of the visual task [26]. The P2 has been discussed to reflect higher order attentional processes and the influence of top-down control, and may also play a role for memory when comparing current sensory input with stored memory. As a correlate of executive control functions, the fronto-central N2 was measured as well, peaking around 200 to 300 ms after stimulus onset. The N2 is most pronounced when a task requires attentional control, e.g. for suppressing irrelevant information [27]. Finally, the P3b was analyzed, which peaks over parietal areas within a wide time window of around 250 to 500 ms, depending on response and task conditions. The P3b is regarded as an indicator of S-R mapping or working memory update [5, 28] and is especially pronounced in oddball paradigms when target stimuli are processed [29, 30].

The main aim of the study was to analyze whether, despite the fact that measurement quality substantially suffers from low trial counts, reliable latency measures are achievable due to the large sample size [23], using fully automated analysis procedures. In a first step, analyses were performed in different trial splits. For the comparison of different latency measurement methods, alternate samples were drawn from the complete sample that comprised 1/2, 1/3, 1/4 or 1/5 of the data (alternate-draw analysis). To this end, all valid trials were divided into small groups in the size of the split. Then, the 1$^{st}$, 2$^{nd}$, etc. trials were averaged. The 1/5 split of the data was selected as the smallest sample because it reflects the same trial counts as in a typical ToT analysis of the task, for which the complete sample is subdivided into 5 successive segments.

Peak latencies and 50% fractional area latencies for all the above-described ERP components were measured both in individual averages and in leave-one-out averages. Measures obtained in the Jackknifed data were corrected as proposed by Smulders [18] before entering them into statistical analyses. Firstly, for all measures, split-reliability was estimated based on pairwise correlation coefficients between all subsamples corrected by the Spearman–Brown prophecy formula [31, 32]. Additionally, internal consistency in form of Cronbach alpha for the smallest split was calculated [33].

The measures outlined above, however, always apply to the entire group tested. They do not allow to identify single subjects with distorted data. In regular (small size) studies, participants with very low signal to noise ratio are often eliminated by visual inspection. This approach is not suitable (and in general problematic when investigator bias should be avoided) in large-scale data since a reliable decision appears to be impossible when hundreds of data sets should be rated without any objective metrics. Luck and co-workers [34] recently proposed a measure to quantify the quality of single participants' data. The Standardized Measurement Error (SME) reflects the expected error of a particular measure in each subject. In this way, participants with low signal to noise ratio may be identified and excluded from the dataset based on an objective metric. Therefore, SMEs were also calculated in the present analyses,

based on the smallest split and used for pruning the dataset by eliminating participants with very high SME from the dataset. The quality criteria mentioned above were the also calculated for the pruned dataset.

For this first step, analyses were performed on alternately drawn sub-samples of the trials for each subject. This approach assumes that both SOA and time on task effects will be equally distributed across sub-samples. Thus, measures should be as similar as possible. When experimental effects are investigated, however, effect sizes of valid effects might be maximized when the most appropriate method of latency measures is applied. Since the effect of time on task has been repeatedly reported for the PVT in response times and other behavioral parameters, we tested all latency measures also for this variable. The main aim of these analyses was to determine the quality of different latency measures in terms of effect sizes and to test whether between-subject effects can be restored when the jackknife procedure is applied and interindividual variance is reconstructed based on the method proposed by Smulders [18].

Here, one incompatibility of the methods described so far has to be mentioned. For applying the restoration of interindividual variance in jackknifed data, Smulders [18] proposes to generate leave-one-out-averages separately for each cell of the data set. That means, data are handled separately for each sub-group and each experimental variable. When sample sizes in subpopulations differ substantially from each other, variance between leave-one-out-averages will also differ because of the differences of Ns entering the averages. Small groups' averages potentially have a lower data quality if compared to large ones. This also influences SME when tested in jackknifed data. Thus, an objective criterion for the exclusion of participants with corrupt data may vary across groups and will therefore fail. When taking the data set as a whole for generating leave-one-out-averages and assigning group memberships thereafter, it is unclear thus far, whether between-group variance also can be restored.

## Methods

### Participants

The total number of participants at time of evaluation was 541. Participants were between 20 and 70 years old (mean age: 43,7 years), had no history of neurological or psychiatric disorders and were of good general physical health. All participants had normal or corrected to normal vision. They provided informed written consent prior to entering the experiment and received 160 € for participating in the entire study which included 2 examination days. The study was performed in the context of an ongoing large-scale study, the Dortmund Vital Study [35]. All participants gave their written informed consent. The study conformed to the Code of Ethics of the World Medical Association (Declaration of Helsinki) and was approved by the local Ethical Committee of the Leibniz Research Centre for Working Environment and Human Factors, Dortmund, Germany on January 25th, 2016.

### Task

**Stimuli and procedure.** A Psychomotor Vigilance Task (PVT) was performed at the beginning of a series of cognitive tasks. Following a neuropsychological assessment (about 1h), participants entered the EEG laboratory where they stayed for about 4 hours. Preparation of EEG caps started between 9.30 and 10.00 am. After electrode placement they entered a sound attenuated dimly lit room and were seated in a comfortable armchair in front of a 32 inch, 1920 × 1080 pixels VSG monitor (Display++ LCD, Cambridge Research Systems) with 100 Hz refreshing rate. Hand grips with a force sensitive top surface were used for recording response behavior. Data from the force sensors were digitized and collected with the same sampling rate as the EEG and stored in the same file.

We used a modified version of the classical PVT [22], intended to increase the sensitivity of the test to subtle changes as they may occur in a putatively well rested and healthy population. White disks (80 cd/m$^2$) were presented on dark grey background (20 cd/m$^2$) on the center of the screen (visual angle: 3˚) for 150 ms. The interval between two subsequent stimuli varied randomly between 2 and 8 seconds in 4 stages (SOAs: 2s, 3s, 5s, 8s). Participants were asked to give a speeded button press with the dominant hand to any stimulus appearing on the screen. The overall task duration was 10 minutes (132 trials, 32 for each SOA).

**Levels of analyses.** The first part of the data presented is based on alternately selected trials (*alternate-draw* analyses). Therefore, the list of correctly responded to and artifact free trials were divided into N/(size of the split) portions. Then the 1$^{st}$, 2$^{nd}$ etc. trials were selected and averaged. This way, the overall sample was divided into 2, 3, 4, or 5 subsamples of equal size (1/2 to 1/5 splits). The smallest draw, that is, the 1/5 split was selected since it corresponds to the number of trials that remains when a common division of PVT-data is made for ToT analyses (see *content-based* analyses).

In the second part, the dataset was divided into 5 successive segments of 2 minutes duration each. Data were averaged for trials within these time segments and analyzed for effects of ToT (see *content-based* analyses). In order to introduce a between-subjects factor, the sample was divided into 3 age groups of approximately equal size. Young participants were between 20 and 35 years old (N = 153; mean age: 27.9 years), middle aged between 36 and 52 years old (N = 158, mean age: 45.1 years), and older participants between 53 and 70 years old (N = 136, mean age: 59.9 years). Thus, data were entered into a mixed-design ANOVA with the between-subject factor Age (3) and the within-subject factor ToT.

## Recording and EEG data processing

**Data recording.** The EEG was recorded from 60 Ag/AgCI active scalp electrodes (Acti-Cap; Brain Products, Gilching, Germany), mounted in an elastic cap according to the extended 10/20 System [36]. The EEG was pre-processed by a BrainAmp DC amplifier (Brain Products, Gilching, Germany) and sampled at a rate of 1000 Hz. An online 250 Hz low-pass filter was applied. The ground electrode was located at Fpz and the online reference at FCz. Impedances of all electrodes were kept below 10 kΩ.

**EEG preprocessing.** Analyses were performed using EEGLAB [37]. For artifact rejection, an established pipeline very similar the approach of Rodrigues et al. [10] was applied. The basic principle of the pipeline was to use traceable and well-documented algorithms that do not require any subjective control or even visual inspection. An additional focus was to optimize the data for short to mid-latency ERPs, requiring that the high pass filter is not higher than 0.1 Hz, which corresponds to a time constant of 0.16s. In particular data decomposition algorithms, underlying artifact removal, such as the independent component analyses (= ICA [38]), are sensitiv to gross artifacts and low frequency activity that should be removed before there are applied to obtain optimal results [39].

Thus, data were initially band-pass filtered between 0.1–40 Hz. Then, to remove gross artifacts, continuous portions of data based on spectrum thresholding were rejected (pop_rej-cont), bad channels were rejected and data were resampled to 250 Hz for subsequent ICA analysis [37, 38]. Data were downsampled in order to reduce processing time. For ICA analysis, data additionally were high-pass filtered at 1 Hz. Epochs from 800 ms preceding target onset to 1600 ms post target onset were extracted for artifact rejection. This unusually long segment length was used to retain the opportunity to also analyze ERPS on the very same dataset which will however, not be reported on here. The time period of 200 ms preceding target onset served as the measurement baseline. Statistics based trial rejection (pop_autorej) was applied

with an amplitude criterion of 500 μV. ICA with varimax rotation was applied on the remaining preprocessed segments. The obtained ICA decomposition was written back to the unsegmented 1000 Hz and 0.1 Hz high pass filtered data where only the segments that contained gross artifacts were excluded. Resulting ICs were categorized by ICLABEL [40]. ICLABEL estimates the contribution of distinct sources (brain, eye, muscle etc.) to extracted independent components (ICs). ICs were removed from the data set when the ICLabel classifier estimated an IC to be eye activity with a probability higher 30% or if it estimated an IC to be brain activity with a probability below 30%. Cleaned data were again segmented as described before and a trial rejection algorithm was applied with the before mentioned parameters.

Participants with less than 7 remaining ICs were removed from the dataset (N = 93).

**Latency measures.** For each ERP component, latency measurements were taken in a predefined time window that was determined by the grand average and parameters that are known from the literature for the ERP component of interest. Based on visual inspection, a gross time window was pre-defined. Within this time window, the latency of the components' maximum was identified. Then, local minima were searched pre and post the peak within the predefined time window. These latencies served as boundaries for subsequent analyses.

A total of four analyses were performed, in which local peak latencies and 50% fractional area latencies were determined (see Fig 1), both with and without applying jackknifing methods.

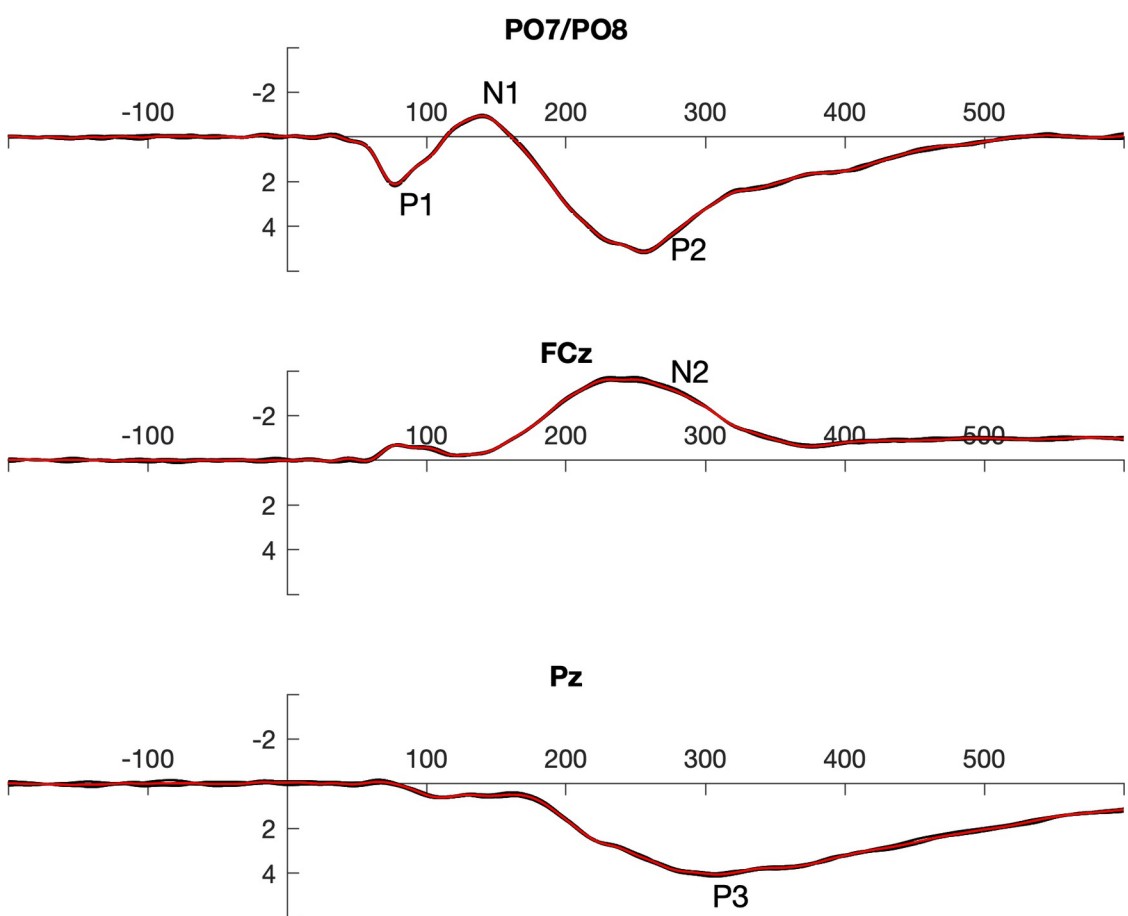

**Fig 2. Grand averages in the 1/5 data split (5 black lines) superposed by the grand-grand average across all subsets (red).** The data demonstrate that there was almost no variance across splits.

Local peaks were defined as the local maximum within the measurement window with the largest positive/negative amplitude.

For 50% fractional latency, half min/max amplitude within the search window was subtracted (see Liesefeld, 2018) to control for low frequency noise and all values below zero were set to zero. Then the time point was determined when the aggregated values crossed the criterion of 50% of the overall aggregate amplitude. The exact latency was determined by linear interpolation between data points.

For Jackknife analyses, N leave-one-out averages were calculated. In this data set local peaks and 50% fractional area latencies were determined as described above for the individual averages. Due to the large size of the data set, the leave-one-out averages were highly similar across participants (see Fig 2). When peaks are determined in such a data set with the given resolution of 1 kHz, outcome of peak latency measurement reveals no variance despite subtle differences in the wave shapes. More than 70% of all measures were identical in the given dataset. In order to increase the temporal resolution of this measure, data were fitted by a Gaussian process regression model as implemented in the Statistics and Machine Learning Toolbox of MATLAB [41] that was applied with a temporal resolution of 1 MHz.

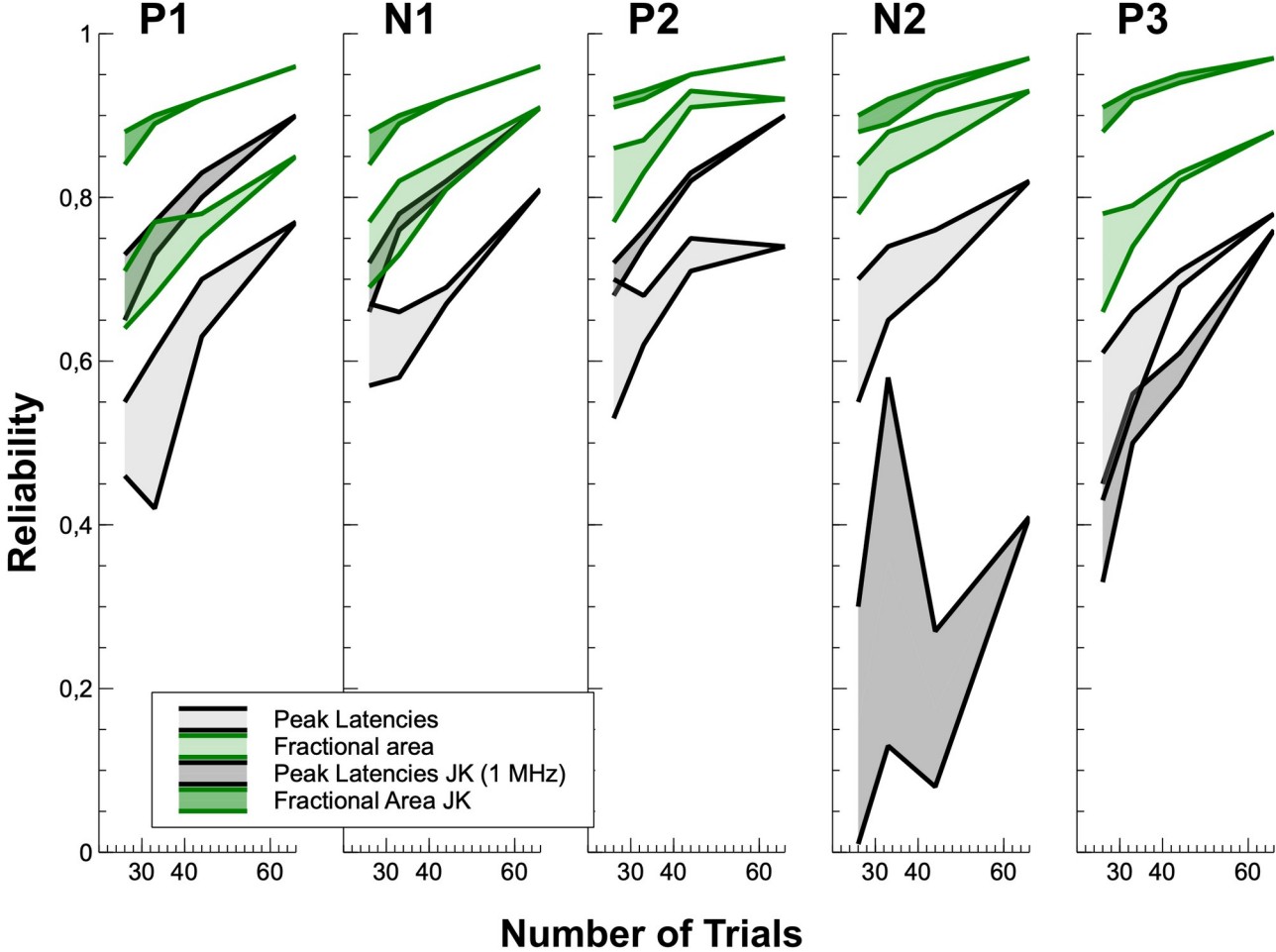

**Fig 3. Spearman-Brown corrected reliability (minimum to maximum) for the five ERP components and the four measures that provided acceptable output for all components.** The four splits were assigned to the number of trials possible, which is defined by the truncated division of the overall number of trials (132) by the size of the split (1/5: 26; 1/4: 33; 1/3: 44; 1/2: 66).

Interindividual variance in all Jackknife measures was reconstructed by the approach as proposed by Smulders [18].

**Components of interest.**   Since a dataset with visual stimulation was investigated, P1 and N1 were evaluated on the average of lateral parieto-occipital electrodes (PO7/PO8) as indicators of early perceptual and attentional mechanisms (see Fig 1). The posterior P2 as indicator for higher order attentional mechanisms was measured at the same electrodes. As indicator of executive control the frontal N2 was measured at FCz. Finally, the parietal P3 was measured at Pz.

**Quality criteria.**   Reliability and internal consistency as quality criteria were investigated in the alternate-draw dataset, because in this dataset, theoretically all measures should be identical when the data would be perfect. For all splits the subsamples were averaged and latency measures were applied. Then correlations between all subsamples were calculated and Spearman-Brown corrected. The highest als lowest correlations were determined and used for data presentation (as the lower and upper limit for reliability respectively). For the smallest subsamples (1/5 data split) additionally Cronbachs Alpha was calculated as a measure of internal consistency.

For each subject the Standardized Measurement Error (= SME [34]) was calculated across the 5 measures in the 1/5 data split for each latency measure. Participants with an SME larger than the mean plus three standard deviations—which is the most common approach towards outlier correction—were excluded from further analyses. This way, a pruned (reduced) dataset was generated and all quality parameters were calculated anew.

**Content-based analyses.**   Vigilance is normally defined not only based on the absolute level in behavioral data (response times and error rates), but also on the decline of these parameters across the test. Thus, in order to test whether the analysis procedures equally reflect the expected ToT effect (in form of an increase in ERP latencies in the course of the task), we applied the common split of the task into 5 subsequent segments of 2-minutes each (time on task; note that the number of trials was comparable to the 1/5 split in the alternate-draw dataset). Data were averaged within this time windows and the three most promising analysis methods as derived from the quality criteria testing were applied. Since we were interested in the applicability of the measures derived also for testing of between subject effects, we divided the sample into 3 age-groups of approximately the same size. Thus, measured latencies were entered into an ANOVA with the between-subject factor age (3) and the within-subject factor ToT (5). This has been done with the original sample as well as the pruned sample that was reduced based on SME (see above). Additionally, 50% fractional area latencies were also measured in leave-one-out-averages that were calculated separately for each age group and tested the same way as the data obtained from the global data set.

## Results

### Consistency of latency measures

The reliability in the alternative draw data (see Fig 3 and S1 Table for minima and maxima for the different splits in the supplementary material) increased as expected with increasing numbers of trials in the average. When the usual criterion for this test (r = .7) was considered, only measures of fractional latency, both in individual and in jackknife data, provided good results across all splits tested. Measurement of local peaks clearly failed with low trial numbers. As was already outlined above, peak picking in jackknife data faced a substantial problem in such a large data set. Since the leave-one-out averages were almost identical with such a large number of subjects, none or only minor variance remained for testing the effects. Overcoming this failure by fitting a Gaussian regression model on the data with a potentiated virtual sampling

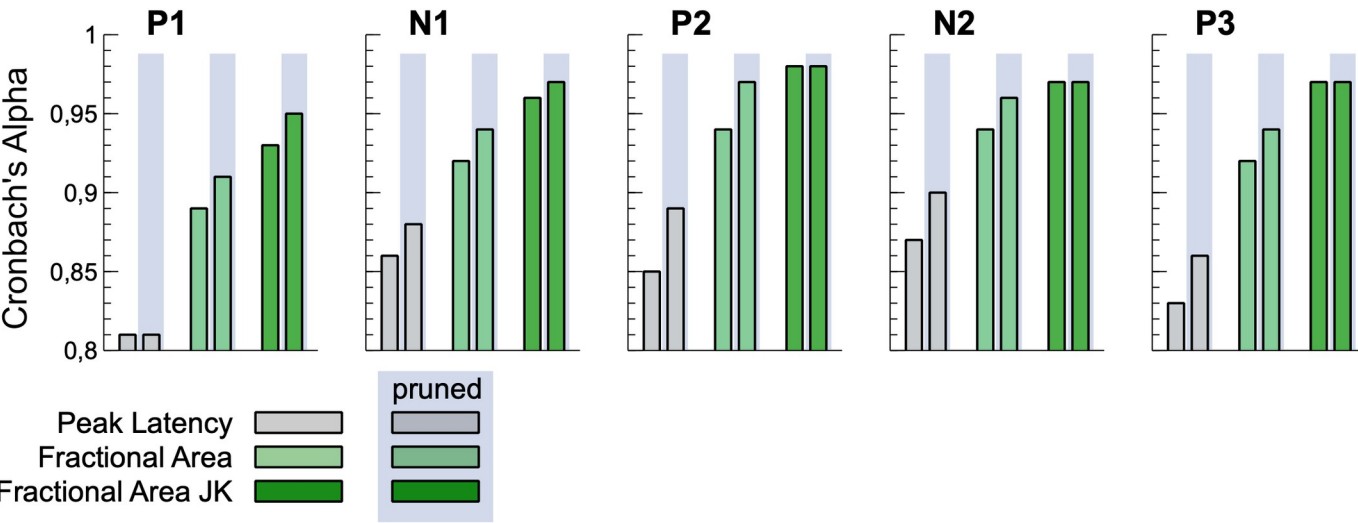

**Fig 4. Cronbach's Alpha in the smallest split for the original data set and the dataset pruned with SME.**

rate (10 MHz) did not work out for all components tested. In particular for late components (N2 and P3), upsampling was not sufficient for restoring variance in the measures.

Overall (see Fig 2), 50% fractional area latency in jackknifed data provided the best results when reliability was considered. This method did not even fail to provide reliable data with very low trial counts (1/5 data split). Additionally, it has to be mentioned that the spread between minimal and maximal reliability obtained within splits was the smallest for this measure, which additionally supports the assumption that data quality was best for the combination of jackknifing and 50% fractional area latency.

Despite the observed spread of reliabilities in the 1/5 data split, Cronbach's Alpha (see Fig 4 and S2 Table) was sufficient for all measures, except the local peaks in jackknifed data. Taking the absolute numbers, again, fractional latency in jackknifed data outperformed all other measures and reached values above .93 for all components measured.

The Standardized Measurement Error (SME) showed the same pattern as Cronbach's Alpha. When participants with an SME larger than 3 standard deviations above mean were excluded (pruned data), internal consistency increased for most measures.

## Content-based analyses

Within-subject effects (see Table 1) strongly varied across methods and with pruning of the data. The local peaks showed smallest effect sizes, followed by 50% fractional area latency. As for the quality testing reported above, fractional area latency in jackknifed data provided the strongest effects. Note that adjusted partial eta squared—as a parameter of effect size—may increase essentially for some measures.

This pattern was visible for the N1, P2 and N2 components. When these effects are inspected in the grand averages (see Fig 5), one may notice that these ERP components unequivocally also show a latency effect in the grand average. The P1 appears to be relatively stable in latency with respect to its peak, but the overall wave-shape is affected by modulations of its offset edge. The P3, on the other hand, appears to be modified in its morphology and amplitude with time on task. Both characteristics led to reliable time on task effects that were captured similarly by all methods.

**Table 1. Outcome of the mixed-design ANOVAS for peak latencies and fractional area latencies in individual averages and in jackknifed data.** Data were reanalyzed after method specific reduction of the dataset based on the Standardized Measurement Error (pruned data = "pr_"). Additionally, analyses were run based on jackknife averages separately for the three age groups (cell-based = "c_"). Number of valid subjects in the analyses are given in brackets for all measures for the first component but apply to all components analyzed.

| | | Age | | | Time on Task | | | Age by Time on Task | | |
|---|---|---|---|---|---|---|---|---|---|---|
| | | **F** | **p** | **adj $\eta_p^2$** | **F** | **p** | **adj $\eta_p^2$** | **F** | **p** | **adj $\eta_p^2$** |
| P1 | Peak (447) | 5.82 | 0.003 | 0.021 | 6.76 | <0.001 | 0.013 | 0.95 | >.2 | 0 |
| | pr_P (409) | 5.63 | 0.004 | 0.022 | 6.11 | <0.001 | 0.012 | 0.55 | >.2 | < 0 |
| | Fract (447) | 3.68 | 0.026 | 0.012 | 5.05 | <0.001 | 0.009 | 1.28 | >.2 | 0.001 |
| | pr_Fr (407) | 3.62 | 0.028 | 0.013 | 6.22 | <0.001 | 0.013 | 1.17 | >.2 | 0 |
| | FractJK (447) | 9.80 | <0.001 | 0.038 | 3.86 | 0.004 | 0.006 | 0.74 | >.2 | < 0 |
| | pr_FrJK (422) | 8.82 | <0.001 | 0.036 | 3.79 | 0.005 | 0.007 | 0.24 | >.2 | < 0 |
| | c_FrJK | 12.88 | <0.001 | 0.051 | 4.07 | 0.003 | 0.007 | 1.01 | >.2 | 0 |
| N1 | Peak | 11.32 | <0.001 | 0.044 | 3.80 | 0.005 | 0.006 | 0.87 | >.2 | < 0 |
| | pr_P | 11.21 | <0.001 | 0.048 | 4.24 | 0.002 | 0.008 | 1.87 | 0.063 | 0.004 |
| | Fract | 12.81 | <0.001 | 0.050 | 11.58 | <0.001 | 0.023 | 0.85 | >.2 | < 0 |
| | pr_Fr | 12.12 | <0.001 | 0.052 | 19.69 | <0.001 | 0.044 | 1.67 | 0.107 | 0.003 |
| | FractJK | 14.24 | <0.001 | 0.056 | 33.46 | <0.001 | 0.068 | 1.28 | >.2 | 0.001 |
| | pr_FrJK | 18.05 | <0.001 | 0.075 | 37.55 | <0.001 | 0.080 | 1.07 | >.2 | 0 |
| | c_FrJK | 17.65 | <0.001 | 0.070 | 28.62 | <0.001 | 0.058 | 1.16 | >.2 | 0.001 |
| P2 | Peak | 0.49 | >.2 | < 0 | 3.65 | 0.007 | 0.006 | 1.44 | 0.180 | 0.002 |
| | pr_P | 0.24 | >.2 | < 0 | 4.92 | 0.001 | 0.010 | 1.90 | 0.062 | 0.004 |
| | Fract | 0.73 | >.2 | < 0 | 8.79 | <0.001 | 0.017 | 2.42 | 0.019 | 0.006 |
| | pr_Fr | 0.70 | >.2 | < 0 | 10.48 | <0.001 | 0.023 | 2.90 | 0.005 | 0.009 |
| | FractJK | 0.13 | >.2 | < 0 | 23.05 | <0.001 | 0.047 | 2.90 | 0.005 | 0.008 |
| | pr_FrJK | 0.29 | >.2 | < 0 | 25.62 | <0.001 | 0.055 | 2.37 | 0.019 | 0.007 |
| | c_FrJK | 0.17 | >.2 | <0 | 25.53 | <0.001 | 0.052 | 2.67 | 0.008 | 0.007 |
| N2 | Peak | 4.59 | 0.011 | 0.016 | 25.01 | <0.001 | 0.051 | 1.12 | >.2 | -< 0 |
| | pr_P | 3.60 | 0.028 | 0.013 | 26.20 | <0.001 | 0.058 | 0.82 | >.2 | < 0 |
| | Fract | 4.23 | 0.015 | 0.014 | 45.75 | <0.001 | 0.091 | 1.23 | >.2 | 0.001 |
| | pr_Fr | 3.37 | 0.035 | 0.012 | 52.83 | <0.001 | 0.114 | 1.07 | >.2 | 0 |
| | FractJK | 2.69 | 0.069 | 0.008 | 58.10 | <0.001 | 0.114 | 1.62 | 0.123 | 0.003 |
| | pr_FrJK | 3.12 | 0.045 | 0.010 | 60.81 | <0.001 | 0.125 | 1.83 | 0.075 | 0.004 |
| | c_FrJK | 3.28 | 0.039 | 0.010 | 54.37 | <0.001 | 0.107 | 1.41 | 0.194 | 0.002 |
| P3 | Peak | 1.87 | 0.155 | 0.004 | 5.80 | <0.001 | 0.011 | 1.37 | >.2 | 0.002 |
| | pr_P | 4.31 | 0.014 | 0.016 | 7.28 | <0.001 | 0.015 | 1.66 | 0.108 | 0.003 |
| | Fract | 1.53 | >.2 | 0.002 | 9.54 | <0.001 | 0.019 | 1.49 | 0.161 | 0.002 |
| | pr_Fr | 1.50 | >.2 | 0.003 | 13.49 | <0.001 | 0.030 | 1.38 | >.2 | 0.002 |
| | FractJK | 1.76 | 0.173 | 0.003 | 7.51 | <0.001 | 0.014 | 0.66 | >.2 | < 0 |
| | pr_FrJK | 2.49 | 0.084 | 0.007 | 7.65 | <0.001 | 0.016 | 0.86 | >.2 | < 0 |
| | c_FrJK | 1.90 | 0.151 | 0.004 | 7.21 | < .001 | 0.014 | 0.68 | >.2 | <0 |

Peak: peak latency; pr_P: peak latency in pruned data; Fract: fractional area latency (50%); pr_Fr: fractional area latency (50%) in pruned data; FractJK: fractional area latency in jackknifed data (50%); pr_FrJK: fractional area latency in jackknifed and pruned data (50%); c_FrJK: cell-based fractional area latency, jackknifed.

Between-subject effects (see Table 1) were not systematically affected by the different measures or pruning based on SME. In other words, effects of Age were significant (or not) largely independent of the method used for data processing. The same held true for the main effect of Time on Task and the interaction of Age by Time on Task. Overall, it must be said that all methods delivered, despite the large methodological differences, quite similar outcomes, also

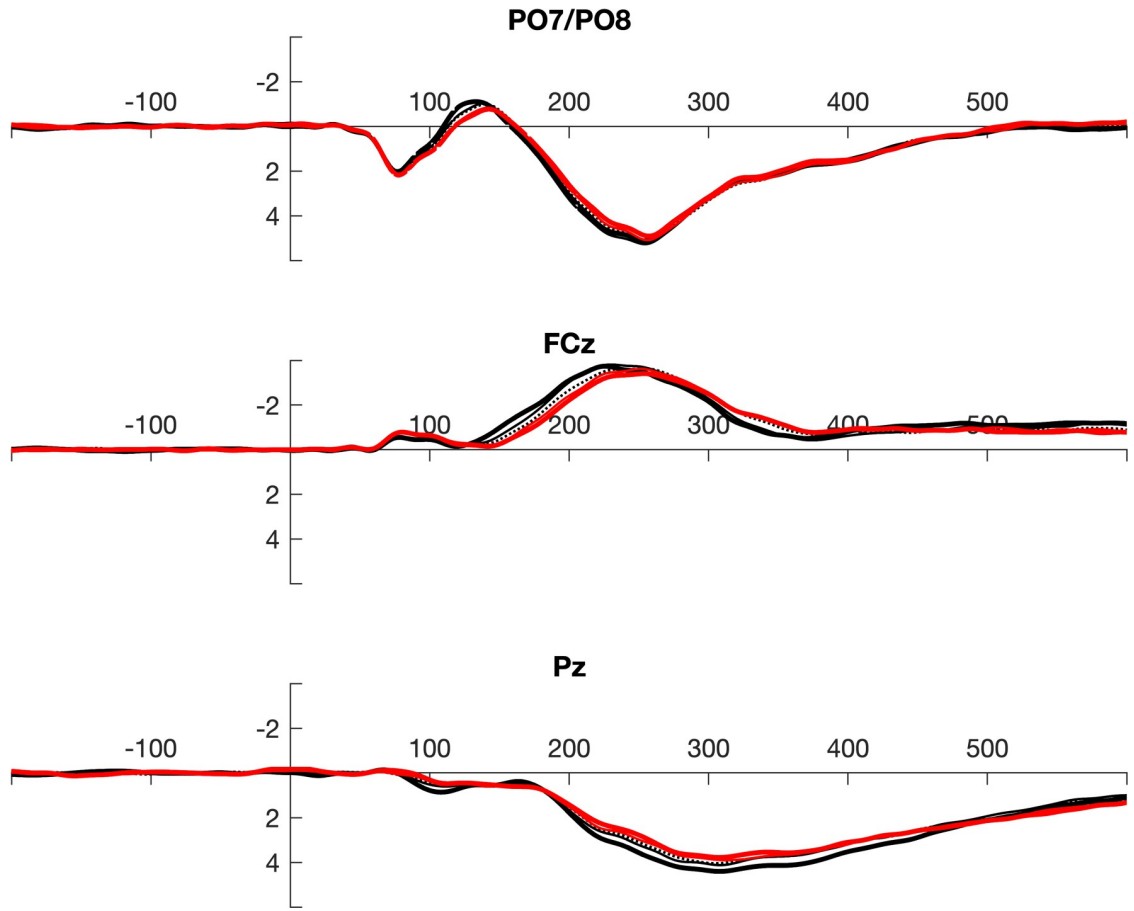

**Fig 5. Grand averages for the 5 time on task segments (1: Black bold, 2: Black thin, 3: Black dotted, 4: Red thin, 5: Red bold).** For N1, N2 and P2 components, latencies increase clearly with time on task (ToT). For the P1 only a modulation in the offset is visible that might be already driven by the following N1 effect. P3 changes before most in amplitude and morphology.

when absolute numbers are considered. All effects of central interest were almost identical when measures from the whole data set were compared with the cell-based analyses (leave-one-out averages are generated separately for age groups). In Fig 6 all mean values with their 95% confidence interval are shown for younger and older participants. Both, level and time course of the effects within the groups were comparable.

## Discussion

One of the core features of event-related activity in the EEG is its high temporal resolution that allows to estimate the timing of mental processes, also called mental chronometry [1, 3]. Large-scale studies may help to uncover even subtle effects in cognitive processing. However, peak latency measures derived from large-scale datasets, may suffer from noisy signals, both due to the unfeasibility to visually inspect and preprocess the data interactively and also because of the low trial counts resulting from study restrictions. The high number of individual datasets may, on the other hand, at least in part overcome this deficit. Here, we tried to demonstrate how the two most common methods to estimate latencies in ERP data can be applied in such studies and how they perform with respect to well-known quality measures.

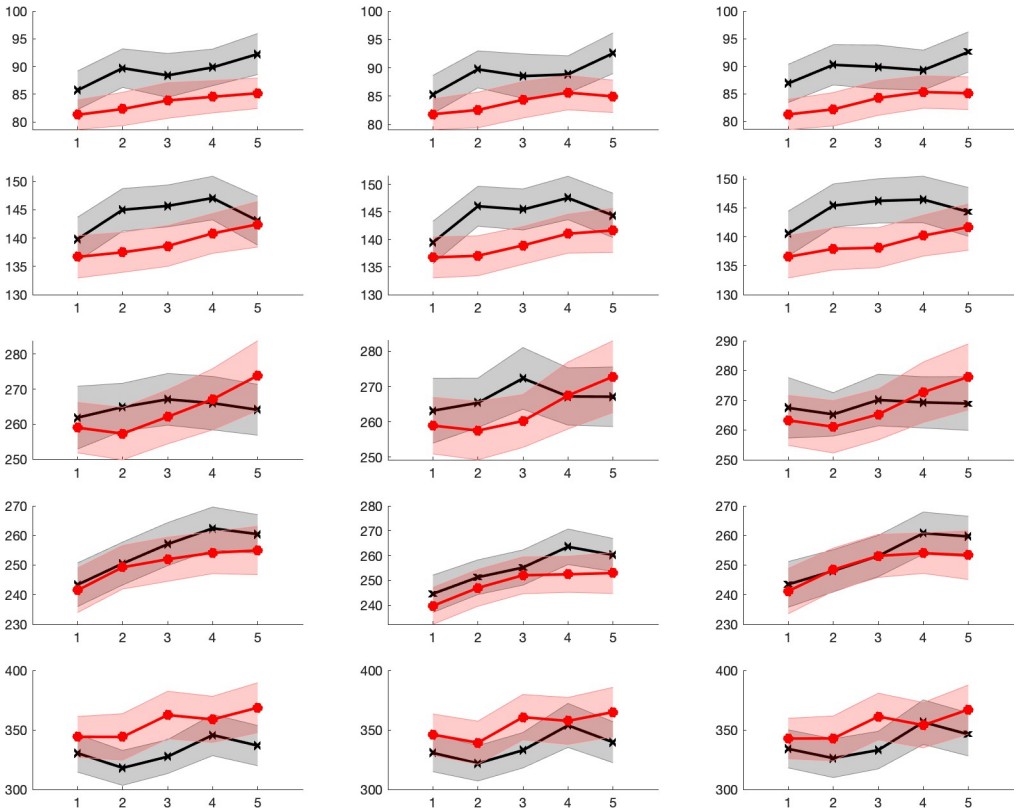

**Fig 6. Means and shaded 95% confidence intervals for local peaks (left column), 50% fractional area latency (middle column) and 50% jackknife fractional area latency (right column) for the 5 time on task segments.** Data are superposed for younger (black) and older (red) participants. Note that the time courses both of within-subject as well as of between-subject effects are highly comparable between measures.

We compared peak latencies and 50% fractional area latencies in individual averages and in jackknifed data. Despite the assumed strong influence of high frequency noise present in the data on peak latencies in individual averages, Cronbach's Alpha was sufficiently high even for this approach, which can be assumed to be most vulnerable to distortion. However, a closer look at split-reliabilities shows that data were in fact quite noisy. Fractional area latency provided, as expected, substantially more reliable data but the range of split-reliabilities was still quite high.

Jackknifing reduced the influence of high frequency noise. However, in such large data sets, the high similarity of the leave-one-out averages might lead to a lack of variance of the measures for valid statistics. In fact, determining peak latencies in the regular temporal resolution failed for all ERP component of interest. Fitting a gaussian process regression model to the data allowed to up-sample the ERP data without altering the underlying wave shape. However, up-sampling to 1 MHz did not really solve the problem of low variance in the data presented here. Fractional area latency on the contrary provided stable data for all ERP components. The linear interpolation applied provided a very high temporal resolution that appears to be necessary for measuring latencies in jackknifed large datasets. For sure, the same could be possible based on the gaussian process regression model, but the limit for efficient up-sampling is not known and processing demands may explode when data sets get larger.

The fact that jackknifing fails for peak picking particularly for late and slow components is not new. Kiesel and co-workers [21] observed the same phenomenon in their simulation

study. Also, they obtained measures of fractional latency by linear interpolation while the peaks were taken based on the temporal resolution of the underlying signal. This shortcoming might be overcome to some extent by the approach applied here. However, fractional area latencies have another advantage that should be mentioned. In the present study we used a 50% criterion in order to be in a similar time window as for the peak picking, but other portions can be used to estimate the onset (e.g. 10%) or offset (90%) of a component. This way, a more finely grained analysis of latency effects would be possible.

Recently, Luck and colleagues [34] proposed a new method to estimate data quality on a single subject level. The Standardized Measurement Error (SME) determines the intraindividual variance of a particular measure. The original method has been developed to be applied to single trial amplitude measures but can also be used for latency measures in sub-samples as presented here. Pruning the data set based on SME substantially improved data quality for all measures. Since this method can be applied completely automatically with mathematically defined criteria, it is a useful additional tool for large data sets as an alternative to a manual pruning based on visual inspection of each individual data set.

Finally, we addressed the implications of these approaches to content-based analyses of the data. Of particular interest were the characteristics of between-subject effects that have never before been analyzed in such large data sets based on jackknifed data. Basically, the jackknifing procedure reduces variance in individual measures. Until Smulders [18] proposed his algorithm to restore the interindividual variance, algorithms that adjust for the variance reduction effect of jackknifing had only been worked out for within-subject effects, while between-subject effects could be demonstrated to also be reliably restored with this method.

However, Smulders [18] proposed to generate leave-one-out averages separately for each cell of analysis, i.e. separate averages are generated for each sub-group and each condition. This approach is fine per definition when sample sizes within the different groups are highly similar and when sample sizes are sufficiently large and thus stable. The internal variance between sub-groups can then be assumed to be similar due to the comparable number of subjects entered and therefore also the impact of each subject upon the leave-one-out average. However, when thinking about studies in neurogenetics, where sub-groups might be substantially different in size, measurements based on the whole dataset would be preferable. As was demonstrated in the data presented here, between-subjects effects do not substantially differ between these approaches. Moreover, generating the leave-one-out averages based on the whole data set additionally allows the reliable application of SMEs for pruning the data set.

To sum up, measuring latencies in large scale data may suffer from low signal-to-noise ratio due to restrictions in data handling and study design. Estimating latency measures in single-subject averages is strongly affected by low and high frequency noise and will lead to distorted peak latency measures. Fractional area latency may overcome this deficit but is still prone to noise when trial counts are low. Thus, noise reduction approaches such as jackknifing may substantially improve data quality and stability of statistical effects. We demonstrated that within-subject effect sizes may increase strongly when fractional area latency is measured in jackknife data compared to local peak latencies in individual averages. Between-subject effects were comparable across all methods, indicating that jackknifing does not corrupt or compromise them. Pruning the data by Standardized Measurement Error (SME) additionally helps to improve the data quality in large datasets when temporal measures are considered.

## Supporting information

**S1 Table. Split-reliabilities (Spearman-Brown corrected) for peak latencies, fractional area latencies in individual averages and in jackknifed data.** Additionally, data fitted with a

Gaussian process regression model, upsampled to 1 MHz (Peak Fitted) are presented. Data are presented for all ERP components of interest and for 1/2, 1/3, 1/4, and 1/5 splits of the alternate-draw data set.
(DOCX)

**S2 Table. Internal consistency (Cronbachs Alpha), data quality (Standardized Measurement Error = SME), and the effects of data pruning for the 1/5 data split for all measures retrieved and all ERP components of interest.**
(DOCX)

## Author Contributions

**Conceptualization:** Edmund Wascher, Stephan Getzmann, Stefan Arnau.

**Data curation:** Marie Gutberlet.

**Formal analysis:** Edmund Wascher.

**Investigation:** Edmund Wascher, Stephan Getzmann, Stefan Arnau.

**Methodology:** Edmund Wascher, Fariba Sharifian, Marie Gutberlet, Daniel Schneider, Stephan Getzmann, Stefan Arnau.

**Project administration:** Stephan Getzmann.

**Software:** Fariba Sharifian, Stefan Arnau.

**Validation:** Marie Gutberlet, Daniel Schneider, Stefan Arnau.

**Visualization:** Edmund Wascher.

**Writing – original draft:** Edmund Wascher, Marie Gutberlet, Daniel Schneider, Stephan Getzmann, Stefan Arnau.

**Writing – review & editing:** Edmund Wascher, Marie Gutberlet, Daniel Schneider, Stephan Getzmann, Stefan Arnau.

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
