## [Decision Letter · Decision Letter 0]

7 Mar 2022

PONE-D-21-37772Mental chronometry in big noisy dataPLOS ONE

Dear Dr. Edmund Wascher,

Thank you for submitting your manuscript to PLOS ONE. After careful consideration, we feel that it has merit but does not fully meet PLOS ONE’s publication criteria as it currently stands. Therefore, we invite you to submit a revised version of the manuscript that addresses the points raised during the review process.

We look forward to receiving your revised manuscript.

Kind regards,

Wajid Mumtaz

Academic Editor

PLOS ONE

Journal Requirements:

Additional Editor Comments (if provided):

Reviewers' comments:

Reviewer's Responses to Questions

**Comments to the Author**

1. Is the manuscript technically sound, and do the data support the conclusions?

Reviewer #1: Yes

2. Has the statistical analysis been performed appropriately and rigorously? 

Reviewer #1: Yes

3. Have the authors made all data underlying the findings in their manuscript fully available?

Reviewer #1: Yes

4. Is the manuscript presented in an intelligible fashion and written in standard English?

Reviewer #1: Yes

5. Review Comments to the Author

Reviewer #1: This is an interesting paper. It deals with important methodological issues and it provides a beautiful overview of some recent approaches to handle big data. The issue of ERPs moreover is a timely topic with a lot of interest for cognitive neuroscience. The whole idea of “mental chronometry”, therefore, is quite challenging. The paper as a whole is well written, with a sound methodology, and some interesting and convincing results, but the readability and understandability of the technical matters could be a problem for the common readership of PLoS One, which consists not only of highly trained neuroscientists. I therefore suggest to add more intuitive descriptions of technical terms and to be more explicit in the description of some methodological procedures. In order to provide a critical-constructive review, I list below some general remarks and detailed comments. I propose to accept the paper for publication on condition that the remarks and comments are addressed appropriately.

General remarks

The English use is OK and is quite idiomatic.

The explanation of the basic concepts of the paper should be explained in more intuitive terms at their first appearance. This can be very short, but it is important that non-informed readers can have an intuition of the meaning of the terms (see below for detailed comments).

Concepts as peak latency, fractional area latency and jackknife procedure need a better intuitive description, given their importance for the whole paper. Even if these are common terms for trained neuroscientists, they are not easily understandable for readers outside of the field. An additional figure to illustrate the intuition behind the terms could also be helpful here.

The methodology seems to be sound and very strict, nut is not easy to understand. Some more elaborate digression in the methodology section should be welcome so as to guide the reader somewhat through the paper.

There are some very interesting results. Perhaps some additional explaining text about the role and measurement of the grand averages could add additional value to these findings and could make the take home message stronger.

The use of quality criteria is not clear and should be explained more in detail.

The paper as a whole is very technical. This is a strength, but somewhat at the cost of understandability and readability. A better description of some technicalities would increase the impact of the paper.

Detailed comments

page 8: there are no keywords

page 9, 1st paragraph: here the concept of peak latency is first introduced. Provide a very short description of the intuitive meaning of the term.

page 9, 2nd paragraph: same remark with respect to the term “connectome”

page 10: same remark with respect to the three focal concepts of peak latencies, fractional area latencies (give the percentages), and jackknifing procedure. Especially the latter is not sufficiently explained. Also the motivation for the N-1 subject average is not sufficiently clear. Please provide the motivation behind this procedure.

page 11, 2nd par.: trial count would never be are than 26: not clear, explain better

page 12,3rd par.: fractional area latencies: explain better the point that divides the area into two regions (equal size: 50 %, or another division). This is given later in the paper, but it should be given here at first appearance. A possible example of a short intuitive description could be: “fractional area latencies measure the area within a time window and finds the time point that divides the area into a specific fraction.” Such short intuitive explanations can do a lot to improve the overall readability of a technical paper.

page 15, 1st par.: explain better how you came to the number of 132 trials.

page 15, 2nd par.: explain a little more in detail what is meant with alternate draw analysis (can be very short). This paragraph as a whole is difficult to understand. Please be clearer here.

page 15, last par.: ICA: I guess this means Independent component analysis? Please write in full with the abbreviation between brackets at first appearance.

page 17, last par.: this whole paragraph about quality criteria is not clear. Explain better what is meant with quality criteria and what the aim is of this paragraph. Explain also how to understand the pruned dataset.

page 18, last par.: explain the concept of split-reliability in more intuitive terms.

page 19: fig. 2: caption: the four splits are not clear. Please explain better, either in the main text or in the figure caption. Try to avoid that readers must invest too much time in deciphering the figures.

page 20: perhaps a stupid question, but how to interpret the F-value of the table. What has been tested? What kind of statistical significance? The table seems to be very rich in content, but it is quite difficult to do the interpretation. Some guidance for the reader should be welcome.

6. PLOS authors have the option to publish the peer review history of their article (what does this mean?). If published, this will include your full peer review and any attached files.

Reviewer #1: No

---

## [Author Response · Author response to Decision Letter 0]

31 Mar 2022

Action letter revision “Mental chronometry in big noisy data”

Please ensure that your manuscript meets PLOS ONEs style requirements

Style has been checked.

References have been corrected. 

We note that you have indicated that data from this study are available upon request. 

Data as requested are now available at https://osf.io/73f8u/

4. Please include your full ethics statement in the ‘Methods’ section of your manuscript file. 

ethics statement is added

5. Please review your reference list 

Done 

Comments to the Author

Review Comments to the Author

Reviewer #1: This is an interesting paper. It deals with important methodological issues and it provides a beautiful overview of some recent approaches to handle big data. The issue of ERPs moreover is a timely topic with a lot of interest for cognitive neuroscience. The whole idea of “mental chronometry”, therefore, is quite challenging. The paper as a whole is well written, with a sound methodology, and some interesting and convincing results, but the readability and understandability of the technical matters could be a problem for the common readership of PLoS One, which consists not only of highly trained neuroscientists. I therefore suggest to add more intuitive descriptions of technical terms and to be more explicit in the description of some methodological procedures. 

General remarks

The English use is OK and is quite idiomatic.

The explanation of the basic concepts of the paper should be explained in more intuitive terms at their first appearance. This can be very short, but it is important that non-informed readers can have an intuition of the meaning of the terms (see below for detailed comments).

Concepts as peak latency, fractional area latency and jackknife procedure need a better intuitive description, given their importance for the whole paper. Even if these are common terms for trained neuroscientists, they are not easily understandable for readers outside of the field. An additional figure to illustrate the intuition behind the terms could also be helpful here.

A new figure 1 is added with the symbolic presentation of peak latency and fractional area latency. Additionally, the problem of high frequency noise is part of the presentation

Figure 1. Simulated data for two subsets of a typical posterior waveshape (black and red lines), reflecting a P3 component. Both, peak latencies and 50% fractional area latencies were extracted. Peak latencies denote a single moment of maximal voltage within the designated measurement window. Fractional area latency denotes the moment when the cumulative area under the curve exceeds a defined criterion, here 50% of the overall area. Despite the high similarity of morphology across the two subsets, peak latencies would show an effect of more than 50 ms due to high frequency noise (solid vertical lines), that did not affect fractional area latency to the same amount (dashed vertical lines). 

Also the jackknifing procedure is described now more extensively.

The methodology seems to be sound and very strict, but is not easy to understand. Some more elaborate digression in the methodology section should be welcome so as to guide the reader somewhat through the paper.

A paragraph was added, describing the rationale for the processing pipeline. Arguments for single processing steps are added.

There are some very interesting results. Perhaps some additional explaining text about the role and measurement of the grand averages could add additional value to these findings and could make the take home message stronger.

We did not really understand this point. The grand average is just a overall average for presentation that is not subject to measurement.

The use of quality criteria is not clear and should be explained more in detail.

See detailed comments

The paper as a whole is very technical. This is a strength, but somewhat at the cost of understandability and readability. A better description of some technicalities would increase the impact of the paper.

We hope that we managed to make descriptions clearer

Detailed comments

page 8: there are no keywords

Keywords are added

page 9, 1st paragraph: here the concept of peak latency is first introduced. Provide a very short description of the intuitive meaning of the term.

We added a better description of the measures used and a conceptual graph that should make the concepts better to understand 

page 9, 2nd paragraph: same remark with respect to the term “connectome”

The “human connectome project” is described now in more detail

page 10: same remark with respect to the three focal concepts of peak latencies, fractional area latencies (give the percentages), and jackknifing procedure. Especially the latter is not sufficiently explained. Also the motivation for the N-1 subject average is not sufficiently clear. Please provide the motivation behind this procedure.

Motivations and descriptions for procedures are added

page 11, 2nd par.: trial count would never be are than 26: not clear, explain better

Is explained now in detail

page 12,3rd par.: fractional area latencies: explain better the point that divides the area into two regions (equal size: 50 %, or another division). This is given later in the paper, but it should be given here at first appearance. A possible example of a short intuitive description could be: “fractional area latencies measure the area within a time window and finds the time point that divides the area into a specific fraction.” Such short intuitive explanations can do a lot to improve the overall readability of a technical paper.

The concept of fractional area latency is described 2 pages before. Now furthermore, an additional figure illustrates it.

page 15, 1st par.: explain better how you came to the number of 132 trials.

Has been made explicit

page 15, 2nd par.: explain a little more in detail what is meant with alternate draw analysis (can be very short). This paragraph as a whole is difficult to understand. Please be clearer here.

Added

page 15, last par.: ICA: I guess this means Independent component analysis? Please write in full with the abbreviation between brackets at first appearance.

Done

page 17, last par.: this whole paragraph about quality criteria is not clear. Explain better what is meant with quality criteria and what the aim is of this paragraph. Explain also how to understand the pruned dataset.

This small chapter is revised 

page 18, last par.: explain the concept of split-reliability in more intuitive terms.

Is now more explicitly named and also described in the method section more explicitly

page 19: fig. 2: caption: the four splits are not clear. Please explain better, either in the main text or in the figure caption. Try to avoid that readers must invest too much time in deciphering the figures.

Caption of fig.2 (which is now figure 3), has been revised accordingly

page 20: perhaps a stupid question, but how to interpret the F-value of the table. What has been tested? What kind of statistical significance? The table seems to be very rich in content, but it is quite difficult to do the interpretation. Some guidance for the reader should be welcome.

Information about statistics is added in the table capture as well as in the method section

While revising your submission, please upload your figure files to the Preflight Analysis and Conversion Engine (PACE) digital diagnostic tool

All figures were tested by PACE

---

## [Decision Letter · Decision Letter 1]

11 May 2022

Mental chronometry in big noisy data

PONE-D-21-37772R1

Dear Dr. Edmund Wascher,

We’re pleased to inform you that your manuscript has been judged scientifically suitable for publication and will be formally accepted for publication once it meets all outstanding technical requirements.

Kind regards,

Wajid Mumtaz

Academic Editor

PLOS ONE

Additional Editor Comments (optional):

Reviewers' comments:

Reviewer's Responses to Questions

**Comments to the Author**

1. If the authors have adequately addressed your comments raised in a previous round of review and you feel that this manuscript is now acceptable for publication, you may indicate that here to bypass the “Comments to the Author” section, enter your conflict of interest statement in the “Confidential to Editor” section, and submit your "Accept" recommendation.

Reviewer #1: All comments have been addressed

2. Is the manuscript technically sound, and do the data support the conclusions?

Reviewer #1: (No Response)

3. Has the statistical analysis been performed appropriately and rigorously? 

Reviewer #1: (No Response)

4. Have the authors made all data underlying the findings in their manuscript fully available?

Reviewer #1: (No Response)

5. Is the manuscript presented in an intelligible fashion and written in standard English?

Reviewer #1: (No Response)

6. Review Comments to the Author

Reviewer #1: (No Response)

7. PLOS authors have the option to publish the peer review history of their article (what does this mean?). If published, this will include your full peer review and any attached files.

Reviewer #1: No

---

## [Editor Report · Acceptance letter]

24 May 2022

PONE-D-21-37772R1 

Mental chronometry in big noisy data 

Dear Dr. Wascher:

I'm pleased to inform you that your manuscript has been deemed suitable for publication in PLOS ONE. Congratulations! Your manuscript is now with our production department. 

Kind regards, 

on behalf of

Dr. Wajid Mumtaz 

Academic Editor

PLOS ONE